# Investigation of the Effect of Double-Filler Atoms on the Thermoelectric Properties of Ce-YbCo_4_Sb_12_

**DOI:** 10.3390/ma16103819

**Published:** 2023-05-18

**Authors:** Nguyen Vu Binh, Nguyen Van Du, Nayoung Lee, Minji Kang, So Hyeon Ryu, Munhwi Lee, Deokcheol Seo, Woo Hyun Nam, Jong Wook Roh, Soonil Lee, Se Yun Kim, Sang-Mo Koo, Weon Ho Shin, Jung Young Cho

**Affiliations:** 1Advanced Materials Convergence R&D Division, Korea Institute of Ceramic Engineering and Technology (KICET), Jinju 52851, Republic of Korea; nguyenvubinh20121995@gmail.com (N.V.B.); ghsl3@naver.com (S.H.R.); whnam@kicet.re.kr (W.H.N.); 2Faculty of Fundamental Sciences, Phenikaa University, Yen Nghia, Ha-Dong District, Hanoi 10000, Vietnam; du.nguyenvan@phenikaa-uni.edu.vn; 3New & Renewable Energy Power Generation Department, Korea South-East Power Co. (KOEN), Yeongcheon 38837, Republic of Korea; mathism@koenergy.kr; 4School of Nano & Materials Science and Engineering, Kyungpook National University, Sangju 37224, Republic of Korea; jw.roh@knu.ac.kr; 5Department of Materials Convergence and System Engineering, School of Materials Science and Engineering, Changwon National University, Changwon 51140, Republic of Korea; leesoonil@changwon.ac.kr; 6Device Research Center, Samsung Electronics, Suwon 16678, Republic of Korea; ksyvip@gmail.com; 7Department of Electronic Materials Engineering, Kwangwoon University, Seoul 01897, Republic of Korea

**Keywords:** thermoelectric, skutterudite, double-filled

## Abstract

Skutterudite compounds have been studied as potential thermoelectric materials due to their high thermoelectric efficiency, which makes them attractive candidates for applications in thermoelectric power generation. In this study, the effects of double-filling on the thermoelectric properties of the Ce_x_Yb_0.2−x_Co_4_Sb_12_ skutterudite material system were investigated through the process of melt spinning and spark plasma sintering (SPS). By replacing Yb with Ce, the carrier concentration was compensated for by the extra electron from Ce donors, leading to optimized electrical conductivity, Seebeck coefficient, and power factor of the Ce_x_Yb_0.2−x_Co_4_Sb_12_ system. However, at high temperatures, the power factor showed a downturn due to bipolar conduction in the intrinsic conduction regime. The lattice thermal conductivity of the Ce_x_Yb_0.2−x_Co_4_Sb_12_ skutterudite system was clearly suppressed in the range between 0.025 and 0.1 for Ce content, due to the introduction of the dual phonon scattering center from Ce and Yb fillers. The highest *ZT* value of 1.15 at 750 K was achieved for the Ce_0.05_Yb_0.15_Co_4_Sb_12_ sample. The thermoelectric properties could be further improved by controlling the secondary phase formation of CoSb_2_ in this double-filled skutterudite system.

## 1. Introduction

Thermoelectricity is one of the outstanding candidates for waste heat recovery, providing an energy solution for collecting waste heat and converting it to electricity. The performance of thermoelectric materials is evaluated by the figure of merit, *ZT*, which is denoted as *ZT* = (*S*^2^*σT*)/*k*, where *S* is the Seebeck coefficient, σ is the electrical conductivity, *k* is the total thermal conductivity composed of lattice and electronic components, and *T* is the absolute temperature. A high *ZT* value can be achieved with high electrical conductivity, a large Seebeck coefficient, and low thermal conductivity [1,2,3]. However, each parameter in the formula above has its own correlations with others in terms of carrier concentration [2,4]. For example, increasing the carrier concentration increases the electrical conductivity but decreases the Seebeck coefficient, and increases the electronic thermal conductivity, resulting in an increase in total thermal conductivity [5]. Hence, optimization of the carrier concentration of TE materials is needed to achieve high thermoelectric efficiency.

Each thermoelectric material can be adapted to different operating temperature ranges and requires good properties, such as high thermal stability, high chemical stability, low toxicity of the elements, raw material availability, and, particularly, high performance for practical applications [6]. Among various promising materials for high thermoelectric energy conversion efficiency, skutterudites (SKD) based on Co_4_Sb_12_ materials are the best candidates for power generation applications operating in the intermediate temperature range; they are well matched to the above requirements, including good mechanical properties [6,7,8,9]. Intrinsic Co_4_Sb_12_-based SKD materials have good electrical properties, such as a high Seebeck coefficient and electrical conductivity, and large carrier mobility [10]. However, the relatively high thermal conductivity of this system results in an overall low thermoelectric performance. Despite this unfavorable feature of thermal conductivity, SKD material systems have an interesting feature; there are two “voids” in their unit cells that can be filled by guest atoms [10]. This affects the charge properties by providing carrier concentration, strongly scattering phonons, and reducing the lattice thermal conductivity [11,12]. The filler atom in the SKD structure not only contributes electrons without changing the band structure [13,14,15], but also significantly reduces the lattice thermal conductivity by loosely binding the atom in the cage, leading to effective phonon scattering [16,17]. A single Yb-filled SKD was found to have an excellent *ZT* of 1.5 at 850 K, and low lattice thermal conductivity (of less than 1.0 Wm^−1^K^−1^) [18]. However, only phonons that have a frequency of resonance close to the rattle frequency will interact and scatter. Therefore, double filling could strongly induce different filler atom vibration frequencies separately, aiming to extend the spectra of resonant phonon scattering [17]. Moreover, Schnelle et al. reported that a lighter atom and smaller ionic radii are more strongly rattled than heavier and large atoms [19]. For enhancing thermoelectric performance, a couple of atoms were inserted into the SKD structure, such as Yb-In (*ZT* = 0.97 at 750 K, and a lowest lattice of 1.7 Wm^−1^K^−1^) [20], Ba-Yb (*ZT* = 1.36 at 800 K, *k*_lat_ = 0.9 Wm^−1^K^−1^) [21], Ca-Yb (*ZT* = 1.03 at 750 K, *k*_lat_ = 0.7 Wm^−1^K^−1^) [16], with promising results. Ce, similarly to Yb, belongs to rare earth metal group and has smaller size of ion, It is lighter than Yb [22], and can be used as a candidate for reducing lattice thermal conductivity through double filling. However, most of these reported experiments use long-lasting heat treatment to synthesize samples. The SKD materials were synthesized using a rapid method that has been published for more than a decade [23,24,25,26,27,28,29], with a shortened processing time of less than 2 days, with a combination of melt spinning and spark plasma sintering [23]. Melt spinning not only saves time, but also refines the nanostructure, which can increase phonon scattering [30,31,32], resulting in a reduction in thermal conductivity and an increase in thermoelectric efficiency. With these practical advantages for synthesizing SKD materials, along with their good thermoelectric properties, their impact on the industry is positive. SKD materials can be used to produce more efficient thermoelectric generators, which can convert waste heat from industrial processes into electricity in the high-temperature range. This can help to reduce the carbon footprint of many industries and make them more sustainable. Besides, replacing Yb with Ce is potentially a commercial option, because Yb is a rare metal (3 ppm), whereas Ce is a relatively abundant element (68 ppm) [33]. Moreover, the price of Ce is lower than that of Yb (Ce: $5000 per ton; Yb: $50,000 per ton) [34]. In this study, the double filling effect was investigated in a series of Ce_x_Yb_0.2−x_Co_4_Sb_12_ (x = 0, 0.025, 0.05, 0.075, 0.1) samples, with the total filling fraction of Ce-Yb fixed at 0.2, using rapid synthesis of melt spinning followed by spark plasma sintering (SPS).

## 2. Materials and Methods

A series of double-filled samples, Ce_x_Yb_0.2−x_Co_4_Sb_12_ (x = 0, 0.025, 0.05, 0.075, 0.1), were prepared using melt spinning followed by a spark plasma sintering method. The raw materials used in this experiment were Ce (Alfa Aesar, ingot, 99.8%), Co (Alfa Aesar, slug, 99.95%), Sb (TTS chemical, slug, 99.9%), and Yb (Alfa Aesar, powder, 99.9%). The chemicals were weighed according to their stoichiometric ratios and placed into carbon-coated quartz tubes, then sealed under a vacuum and subjected to induction melting for 15 min under a power supply of 20 W. The ingots obtained were loaded into cylinder graphite crucibles with an inner diameter of 0.4 mm, for a rapid solidification process (RSP) which was achieved using the melt spinning method under a power supply of 11 kW and a Cu-wheel speed of 2000 rpm (~26.2 m/s) in an argon atmosphere. The ribbons obtained after melt spinning were collected and hand-ground in an argon-filled glovebox to obtain fine powders. The powders were then consolidated using an SPS process at 963 K for 15 min under a pressure of 50 MPa. The relative densities for all samples studied here were measured on the sintered ones, and found to be greater than 95% of the theoretical values.

X-ray diffraction (XRD) data were collected using a Bruker D8 Advance machine (20 kV, 5 mA) with Cu Kα radiation (λ = 1.5418 Å). The microstructure of the sintered samples was analyzed using a scanning electron microscope (SEM, Verios 460 L, FEI, 10 kV for FESEM, 20 kV for EDS). The electrical conductivity and Seebeck coefficient were measured using a commercial ZEM 3 (ULVAC-RIKO) from room temperature to 773 K on rectangular bar-shaped samples with dimensions of approximately 3 mm × 3 mm × 10 mm. The carrier concentration and carrier mobility measurements at room temperature were performed using a Hall measurement system (ResiTest 8400, Tokyo Corporation). The measurements of thermal diffusivity were carried out in a flowing argon atmosphere using laser flash analysis (Laser Flash DLF-1) between 300 K and 773 K, and the thermal conductivity was calculated from the relationship *k* = *αρCp*, where *α* is the thermal diffusivity, *ρ* is the density, and *Cp* is the specific heat capacity. The value of 0.23 Jg^−1^K^−1^ for *Cp* is used to calculate the thermal conductivity for all temperatures due to its weak dependence on temperature [18,35].

## 3. Results and Discussion

The XRD patterns of a series of Ce_x_Yb_0.2−x_Co_4_Sb_12_ (x = 0, 0.025, 0.05, 0.075, and 0.1) samples are displayed in Figure 1a. It is shown that the XRD patterns exhibit good crystalline SKD, with diffraction peaks indexed to the SKD structure corresponding to JCPDS #00-065-1791 (body-centered cubic skutterudite phase, space group *Im*-3). However, small amounts of the secondary phase of CoSb_2_ are clearly observed around 2*θ* = 32.2° for every composition, as shown in Figure 1b. The secondary phase CoSb_2_ is normally formed in skutterudite [23,36,37], especially in melt spinning, because the resulting ribbons contain a mixture of main material and secondary phase [38]. The intensities of the secondary peaks increase with increasing Ce content, and CoSb_2_ secondary peaks can easily be found in the x = 0.1 sample. Focusing on the main SKD peaks (013) around 2*θ* = 31°, there is a shift of the peaks toward a higher 2*θ* angle with increasing Ce content (Figure 1c), which suggests that the lattice constants for Ce_x_Yb_0.2−x_Co_4_Sb_12_ (x = 0, 0.025, 0.05, 0.075, and 0.1) are reduced. To understand the effect of Ce substitution on the reduction of the lattice parameter, the lattice constant was calculated, and the results are displayed as a function of Ce content in Figure 1d. It was found that the lattice constant was reduced from 9.052 Å for sample x = 0, to 9.041 Å for sample x = 0.10 (Table 1). This reduction can be explained by the Ce ion’s preference for a trivalent state, whereas Yb prefers to be divalent in the skutterudite structure [39,40,41]. Thus, the ionic radius of Ce^3+^ (103 pm) is smaller than that of Yb^2+^ (113 pm) [22]. Therefore, by replacing Yb atoms with Ce substitution atoms, the lattice constant could be decreased, and follows Vegard’s law in all samples, as shown in Figure 1d.

The FE-SEM images of the cross-section surface of SPSed bulk samples Ce_x_Yb_0.2−x_Co_4_Sb_12_ (x = 0, 0.025, 0.05, 0.075, and 0.1) are shown in Figure 2. As shown in Figure 2a–e, the samples with x = 0~0.075 have grain sizes varying between 1 μm and 3 µm. However, in the samples with x = 0.1, as seen in Figure 2e, there are small grains with sizes below 1 μm. During the sintering process under high pressure and temperature, the grains start to grow due to the mobility of atoms at high temperatures, leading to an increase in the size of the grain, which is called grain growth. However, the highest content of Ce (x = 0.1) has a large amount of CoSb_2_ secondary phase that hinders the grain growth of Co_4_Sb_12_ grain. Furthermore, the difference between the samples could also be due to the variation in the microstructure of the ribbons from the RSP. To confirm this, the ribbons of samples Ce = 0.025 and 0.1 were investigated using FE-SEM. In the RSP, the melted composite was pulled out, and contacted the rotating copper wheel to form ribbons. Figure 3a–f show the cross-section of the ribbons from the RSP, and Figure 3a,d show that the thickness of the ribbons is ~12 µm. As shown, the grain size in the surface contacting the copper wheel (Figure 3b,e) is smaller than the one on the opposite surface (Figure 3c,f) (free surface). Based on the detailed contact and free surfaces of each sample, the particle size of the contacted face is around 50 nm, which is much smaller than the size of the free surface at 200 nm. The difference in the size of the particles between the two surfaces can be explained by the different cooling rates. Specifically, on the contacted surfaces, the particles are quickly cooled, and small particles are formed. The free surface has more time to cool, and the particles belonging to these areas are bigger because of the longer growth time. The distribution of each element of sample Ce_0.025_Yb_0.175_Co_4_Sb_12_ was determined by EDS mapping, as shown in Figure 4. The Sb and Co elements are mainly distributed throughout the surface, not only in this sample, but also in others. The Yb and Ce, which have low amounts of filling, are discretely distributed in all samples. The details for actual compositions and oxygen contents are displayed in Appendix A.

The carrier concentration and mobility are shown in Figure 5a as a function of Ce content at room temperature. The strong tendencies of the carrier concentration and mobility have been shown in the Ce_x_Yb_0.2−x_Co_4_Sb_12_ (x = 0, 0.025, 0.05, 0.075, and 0.1) system. It is shown that the carrier concentration increases (from 1.42 × 10^20^ cm^−3^ for Ce = 0.000 sample to ~3.41 × 10^20^ cm^−3^ for Ce = 0.100 sample) as Ce substitution increases. This suggests that Ce atoms also act as donor dopants. Although Ce and Yb are from the same lanthanide family, they have different valence states of ions in the skutterudite matrix. Thus, Ce fillers will donate more electrons to the matrix than Yb fillers, leading to a gradual increase in carrier concentration. Furthermore, the carrier mobility shows the opposite tendency to the carrier concentration. Adding more Ce content increases the carrier concentration, which gives electrons a greater opportunity to interact with ions. This decreases the mean free path, and thus decreases mobility. The electrical conductivity is the combination of carrier concentration and mobility, as shown in Figure 5b. Overall, the electrical conductivity shows a trend of increasing with increasing Ce fillers, and the electrical conductivity decreases at higher temperatures, indicating the characteristics of a heavily doped semiconductor. The unfilled Ce shows a decrease in its electrical conductivity value, from 2085 S/cm at room temperature to 1185 S/cm at 860 K. These trends are also observed in all other Ce-filled samples.

The temperature-dependent Seebeck coefficient and the power factor for Ce_x_Yb_0.2−x_Co_4_Sb_12_ (x = 0, 0.025, 0.05, 0.075, and 0.1) are shown in Figure 6. The Seebeck coefficients for all samples in Figure 6a display negative values, indicating the n-type behavior of the semiconductor. The absolute value of the Seebeck coefficient clearly decreases from sample Ce_0.025_Yb_0.175_Co_4_Sb_12_ (−141.8 µV/K at 300 K) to sample Ce_0.1_Yb_0.1_Co_4_Sb_12_ (−123.2 µV/K at 300 K) as the carrier concentration increases. The power factor shows a trend of decreasing with more Ce content due to the changing electrical conductivity and Seebeck coefficient, as shown in Figure 6b. The highest power factor for Ce-Yb double filler is 50.6 × 10^−4^ W/mK^2^ for sample x = 0.050 at around 715 K. However, at high temperatures above 800 K, the power factor tends to decrease because of the reduction in electrical conductivity. This can be explained by the fact that at high temperatures, the intrinsic carrier is dominantly excited. The high-power factor (around 50 × 10^−4^ W/mK^2^) is obtained for samples Ce = 0 and 0.025.

The Seebeck coefficient can also be determined as a function of carrier concentration and effective mass using the Pisarenko relation for degenerate semiconductors, which can be expressed as the equation below. For this analysis, we assume a single parabolic band and an energy-independent carrier scattering approximation for degenerate semiconductors [19,42]:(1)$$S=\frac{8\pi^{2}k_{B}^{2}T}{3qh^{2}}m^{*}{\left(\frac{\pi }{3n}\right)}^{2/3}$$
where *k_B_* is the Boltzmann constant, *T* is the absolute temperature, *h* is Planck’s constant, *q* is the unit charge of the electron, and *m** is the effective mass. The effective mass *m**/*m*_0_ is calculated and listed in Table 1, and Figure 7 shows the relation between the effective mass and carrier concentration at room temperature. In detail, the effective mass has large values that are 1.77 times the free electron mass value m0 for the sample without Ce concentration, and it then increases by 2.77 times, with the index 0.025 of Ce, and reaches 2.95 for the x = 0.100 sample. This proves that the participation of Ce contributes to the large mass effect in the system. The enhancement of effective mass in this system is associated with the valence instability of the 4f electrons in Ce and Yb-based compounds, and the heaviness of charge carriers has been reported in several cases [43,44,45,46].

The temperature-dependent thermal properties of a series of Ce and Yb double-filled samples are displayed in Figure 8a. The total thermal conductivity decreases with an increase in temperature, indicating that phonon–phonon scattering is dominant. In contrast, as the measured temperature increases further, the total thermal conductivity tends to increase due to the contribution of electron–phonon interaction. To explore the effect of Ce-Yb on the lattice’s thermal conductivity, the electronic thermal conductivity was determined using the Wiedemann–Franz relation (*k_e_* = *LσT*), with the Lorenz number, *L*, calculated using the Seebeck coefficient, following the formula *L* = 1.5 + Exp[−|S|/116] (Appendix A) [47]. The lattice thermal conductivity can be determined by *k*_lat_ = *k*_total_ − *k_e_*, and the result is shown in Figure 8b. The participation of Ce leads to a decrease in lattice thermal conductivity. However, the lattice thermal conductivity of sample 0.100 jumps up to a higher value, which is related to the high intensity of the secondary phase in the XRD result. That is, one would expect the reduction of the lattice thermal conductivities in Ce-Yb double-filled samples compared to the Yb single-filled sample. However, as we have clearly seen in the XRD patterns of our samples, the secondary phase of CoSb_2_, with high lattice thermal conductivity [48], exists in our double-filled samples, resulting in higher values of lattice thermal conductivity compared to the Yb single-filled sample.

The dimensionless figure of merit *ZT* was calculated for all Ce_x_Yb_0.2−x_Co_4_Sb_12_ samples, and the results are shown in Figure 9. The lowest *ZT* was observed for sample x = 0.075, which is related to the lowest power factor and highest total thermal conductivity. The maximum *ZT* value for the double-filled samples in this study was achieved for sample x = 0.05 at 750 K with a value of 1.15, and this is due to the lowest lattice conductivity and highest power factor. This value is higher than that of double-filled Yb-In (*ZT* = 0.97) [20] and Yb-Ca (*ZT* = 1.03) [16] at the same temperature of 750 K, but lower than that of Yb-Ba (*ZT* = 1.36) [21]. This observation can be explained by the fact that the combination of heavier fillers more effectively scatters lattice phonons compared to the combination of lighter fillers in the SKD materials system, resulting in a greater reduction of *k*_lat_ and a higher *ZT* value.

## 4. Conclusions

A small amount of Ce and Yb was successfully introduced into the SKD structure through the process of melt spinning and SPS. XRD data revealed a gradual reduction in the lattice parameter of SKD with increasing Ce contents. The difference in grain size observed in the SEM images is attributed to the cooling rate between the two faces of the RSP ribbons. By replacing Yb atoms with Ce, the deficiency of the carrier concentration was compensated for by the extra electron from the Ce donors. This led to the optimization of the electrical conductivity, Seebeck coefficient, and power factor of the Ce_x_Yb_0.2−x_Co_4_Sb_12_ system. At high temperatures, the power factor shows a downturn due to bipolar conduction in the intrinsic conduction regime. Moreover, the lattice thermal conductivity of the Ce_x_Yb_0.2−x_Co_4_Sb_12_ skutterudite system in the range of 0.025 to 0.1 Ce content was clearly suppressed, as the Ce content increased due to the introduction of the dual phonon scattering center from the Ce and Yb fillers. The highest *ZT* value for the double-filled skutterudite was achieved with a value of 1.15 at 750 K for the Ce_0.05_Yb_0.15_Co_4_Sb_12_ sample. However, this value is lower than that of the Yb single-filled SKD, which could be due to the deterioration effect from the secondary phase of CoSb_2_ in our double-filled SKD system. One would expect an enhancement of thermoelectric performance of the double-filled SKD system, under the combination of the rapid synthesis of melt spinning with the spark plasma sintering process, if the secondary phase formation is well controlled.

## Figures and Tables

**Figure 1 materials-16-03819-f001:**
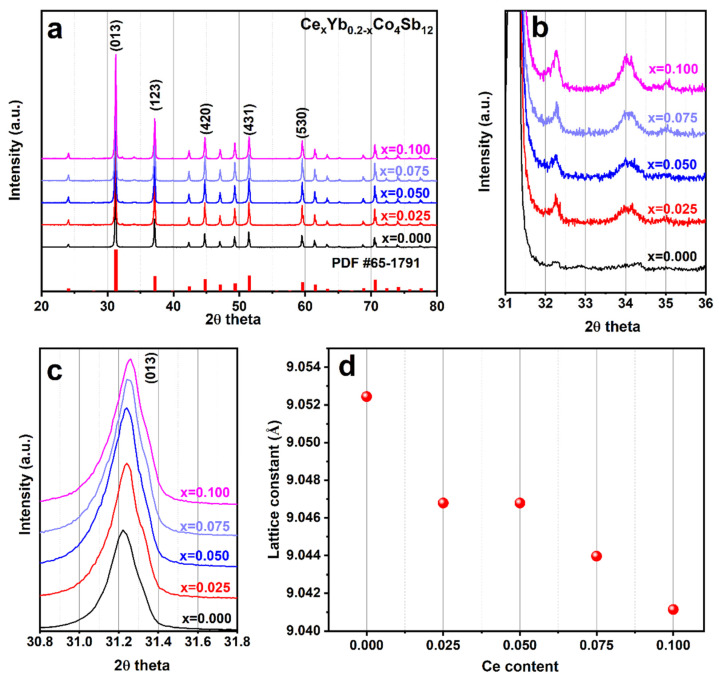
(**a**) Powder XRD of Ce_x_Yb_0.2−x_Co_4_Sb_12_ after SPS, (**b**) the secondary phase peak of CoSb_2_, (**c**) zoomed in XRD pattern between 31° and 31.8°, and (**d**) calculated lattice parameter of Ce_x_Yb_0.2−x_Co_4_Sb_12_ samples as a function of Ce content.

**Figure 2 materials-16-03819-f002:**
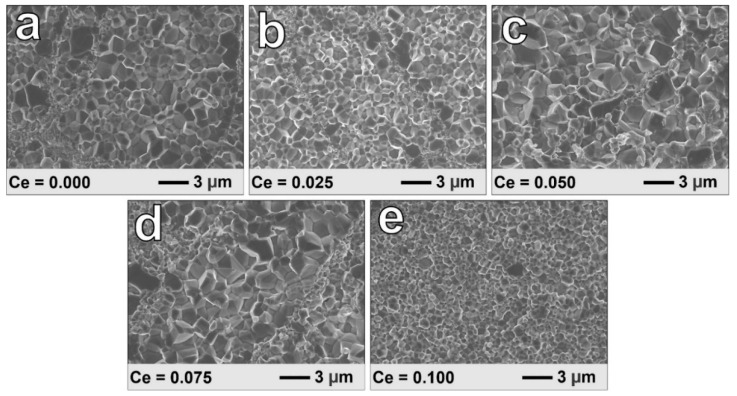
FE-SEM images of Ce_x_Yb_0.2−x_Co_4_Sb_12_ samples after SPS for (**a**) x = 0.000, (**b**) x = 0.025, (**c**) x = 0.050, (**d**) x = 0.075, and (**e**) x = 0.100.

**Figure 3 materials-16-03819-f003:**
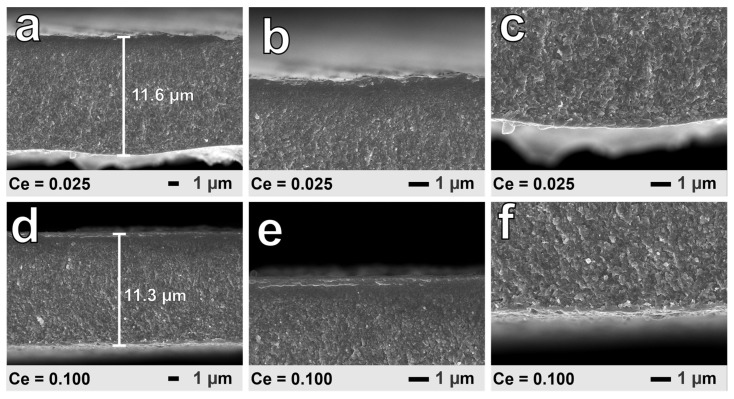
FE-SEM images of ribbons from the RSP process. (**a**) The cross section, (**b**) contact surface, and (**c**) free surface of sample Ce_0.025_Yb_0.175_Co_4_Sb_12_; (**d**) the cross section, (**e**) contact, and (**f**) free surface of sample Ce_0.1_Yb_0.1_Co_4_Sb_12_.

**Figure 4 materials-16-03819-f004:**
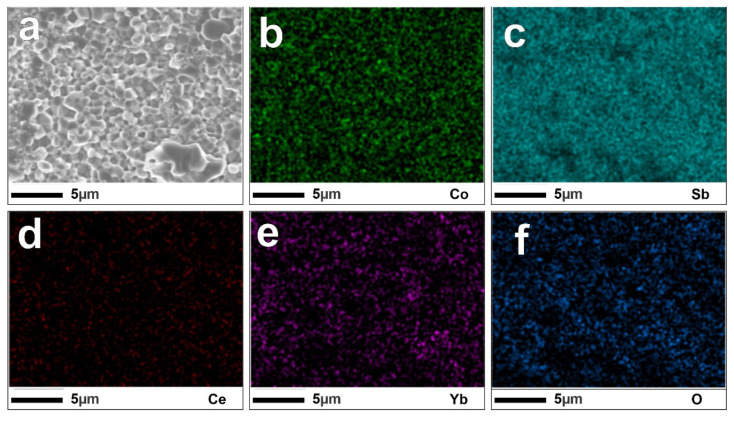
EDS images of sample Ce_0.025_Yb_0.175_Co_4_Sb_12_. (**a**) The SEM image, the distribution of (**b**) Co, (**c**) Sb, (**d**) Ce, (**e**) Yb and (**f**) O (EDS images of other compositions are displayed in Appendix A).

**Figure 5 materials-16-03819-f005:**
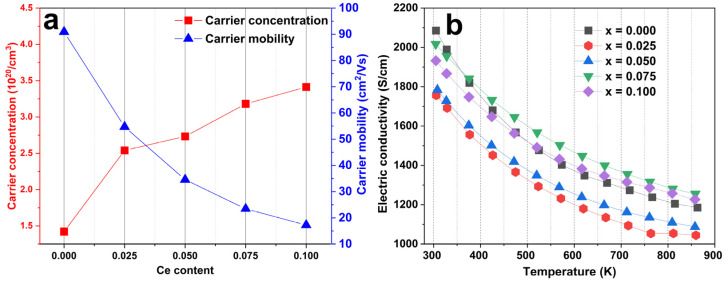
(**a**) The carrier concentration and mobility of Ce_x_Yb_0.2−x_Co_4_Sb_12_ (x = 0, 0.025, 0.05, 0.075, and 0.1) at room temperature, and (**b**) the temperature dependence of electrical conductivity.

**Figure 6 materials-16-03819-f006:**
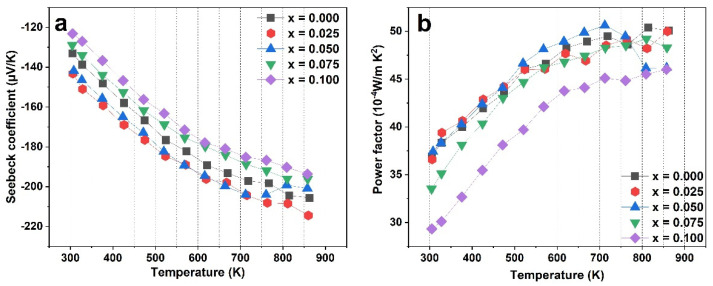
The temperature dependence of (**a**) the Seebeck coefficient and (**b**) the power factor of Ce_x_Yb_0.2−x_Co_4_Sb_12_ (x = 0, 0.025, 0.05, 0.075, and 0.1).

**Figure 7 materials-16-03819-f007:**
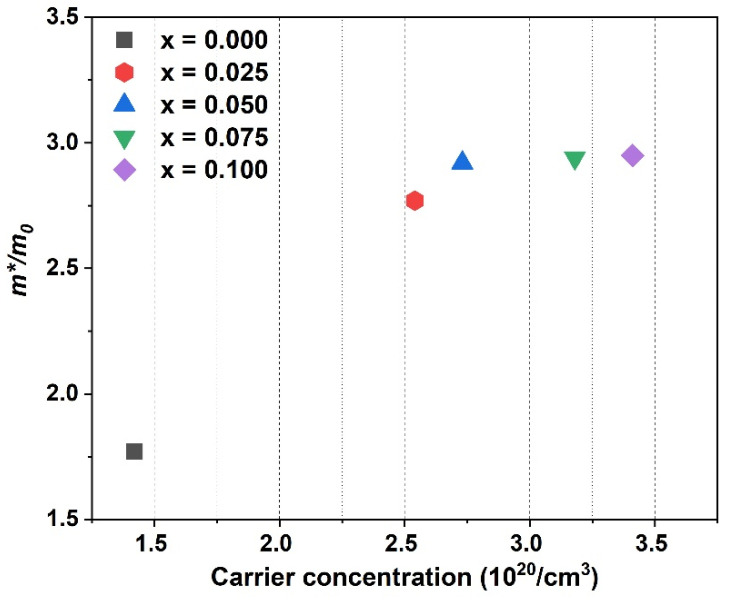
The effective mass *m**/*m*_0_ at room temperature as a function of carrier concentration for Ce_x_Yb_0.2−x_Co_4_Sb_12_.

**Figure 8 materials-16-03819-f008:**
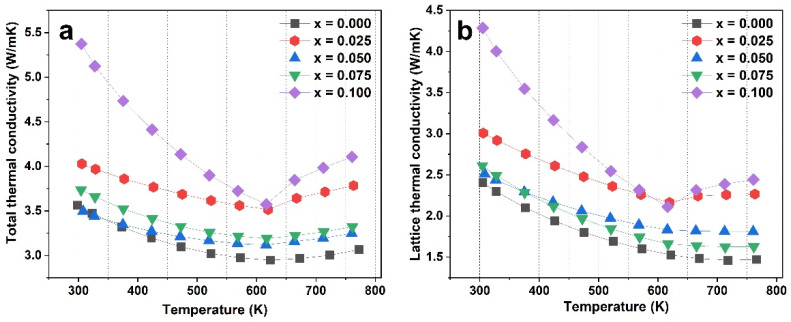
The temperature dependence of (**a**) the total thermal conductivity and (**b**) the lattice thermal conductivity of sample Ce_x_Yb_0.2−x_Co_4_Sb_12_ (x = 0, 0.025, 0.05, 0.075, and 0.1).

**Figure 9 materials-16-03819-f009:**
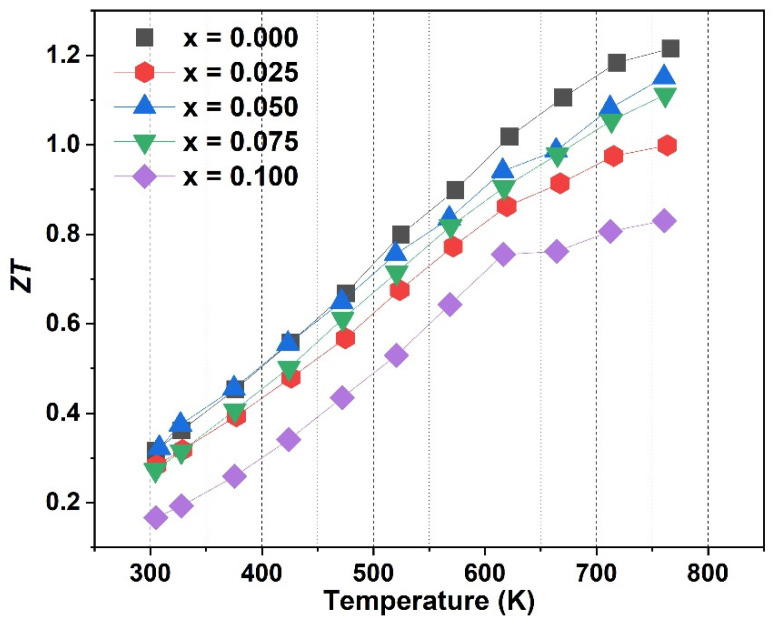
The temperature dependence of the *ZT* of sample Ce_x_Yb_0.2−x_Co_4_Sb_12_ (x = 0, 0.025, 0.05, 0.075, and 0.1).

**Table 1 materials-16-03819-t001:** Nominal composition and room temperature values of the lattice parameter (*a*), carrier concentration (*n*), electrical conductivity (*σ*), Seebeck coefficient (*S*), thermal conductivity (*k*) and effective mass (*m**/*m*_0_) for Ce_x_Yb_0.2−x_Co_4_Sb_12_.

Composition	*a* (Å)	*n* (×10^20^ cm^−3^)	*σ* (Sm^−1^)	S (*µ*VK^−1^)	*k* (Wm^−1^K^−1^)	*m*/m* _0_
Yb_0.2_Co_4_Sb_12_	9.052	1.4	2085.5	−133.1	2.4	1.77
Ce_0.025_Yb_0.175_Co_4_Sb_12_	9.047	2.5	1756.1	−143.2	3.0	2.77
Ce_0.05_Yb_0.15_Co_4_Sb_12_	9.047	2.7	1783.9	−141.8	2.5	2.92
Ce_0.075_Yb_0.125_Co_4_Sb_12_	9.044	3.2	2016.9	−128.9	2.6	2.94
Ce_0.1_Yb_0.1_Co_4_Sb_12_	9.041	3.4	1932.2	−123.2	4.3	2.95

## Data Availability

The data presented in this study are available upon request from the corresponding author.

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
