# Peer review of "Investigation of the Effect of Double-Filler Atoms on the Thermoelectric Properties of Ce-YbCo4Sb12"

_materials, 2023, doi:10.3390/ma16103819_

Round 1

Reviewer 1 Report

Investigation for the effect of double-filler atom on the thermo-electric properties of Ce-YbCo4Sb12

1.       Line 120-121 “Therefore, by replacing Yb atoms with Ce substitution atoms, the lattice constant could be decreased and follows a Vegard’s law in all samples, as shown in Fig. 1d.”  Is it the only reason for the decrease in the lattice constant? Please illustrate how Vegard’s law is satisfied by all the samples.

2.       Table 1 is missing

3.       Fullform of  RSP is missing in the text, Please verify some grammatical errors.

4.       Line 132, “As shown in Fig. 2a-e, ………………small grains with sizes below 1 μm.” How was the grain size confirmed by the author, Please provide more insight to it?

5.       Line 134 “During the sintering process under high pressure and temperature, the grains in the necking area” How is the necking area defined in the grain growth process? What is the significance of this necking area in your case? Does it help in improving the power factor or so?

6.       Line 143 “Fig. 3a and d show that the thickness of the ribbons is ~12 μm” but the thickness is not confirmed from the figures. However the thickness of the samples are shown to be the same for all the samples, in that case the grain size which is 3μm may have more thickness than the one in 1μm for the same scale. 

7.       Line 169, “It is shown that the carrier concentration increases (from 1.42 * 1020 cm-3 for Ce = 0.000 sample to ~3.41 * 1020 cm-3 for Ce = 0.100 sample)” The carrier concentration for Ce = 0.025 sample may be found as the comparison for 0.025 and 0.100 content is under study mostly.

8.       Figure 6b has a power factor as (10-4 W/mK2) however in the text in line 196 and 200 the unit is only W/mK2. Please verify

9.       The figure names and figure captions donot match in figure 8. Please verify

10.   Line 230, “To explore the ………. ……….., L=1.5+Exp[-|S|/116] [26].” What are the values of Seebeck coefficient and electronic thermal conductivity calculated and as the thermal conductivity values are changing at different temperatures means the electronic thermal conductivity values are also changing wrt temperatures. The author could provide this data for clarity.

Need major revision

Reviewer 2 Report

Please see the enclosed comment file.

No issues.

Reviewer 3 Report

The English language and style in the article seem to be fine, and only minor spell checks may be required. The text is clear and comprehensible, with no major issues affecting the understanding of the research presented.

Reviewer 4 Report

Binh and coworkers have explored the transport properties of thermoelectric YbCo4Sb12 doped with Ce. This skutterudite compound is shown to exhibit the figure of merit of nearly 1.2 at 750 K, which is interesting for applications. The authors have utilized the double-filling effect to enhance the thermoelectric performance. At elevated temperatures, they identify the occurrence of the bipolar effect, which is not beneficial, but it is important for the overall understanding. This is surely an interesting work, but there are issues that need to be addressed, as listed below.

1) The authors have not explicitly measured the composition. This is the major shortcoming of the manuscript. How do we know that for instance 0.1 at.% of oxygen is not detrimental for the properties? Oxygen can easily be incorporated during synthesis, while other elements may deviate from the nominal composition based on the precursor materials. EDS mapping (Fig. 4) was used (without introducing the method and conditions) so it would be straightforward to supply the composition.

2)  Another important drawback of the manuscript is that comparison with literature is missing. First, the authors should provide more details in the introduction (ZT values for common skutterudites and other thermoelectric materials). Second, the authors should compare their data throughout the manuscript with the available literature data on skutterudites.

3) The authors have not discussed the influence of stress on the transport properties, but this has been shown to be of relevance (see Acta Mater. 61, 6778 (2013)). Please do not ignore relevant papers.

4) Thermomechanical response of thermoelectric devices is important for their performance since thermal stress and thermal fatigue can easily lead to failure. This issue has been discussed for skutterudites (see Acta Mater. 117, 13 (2016) and Appl. Phys. Lett. 109, 223903 (2016)). It would be beneficial to tackle it in the introduction. A more general view on skutterudites is always welcome and it may attract a larger audience.

5) Details are missing in the methodological section, e.g., power setting for XRD, acceleration voltage for SEM, specific heat capacity value for laser flash, etc. It is important to supply enough details so that the results can be reproduced.

6) There is an error on page 1 regarding the symbol for the thermal conductivity. It is either k or Greek kappa, but it should be not mixed. Please check throughout.

7) The general motivation in the introduction is not clear. What is unknown? What are open questions? Why is it important to carry out this work? Why should Ce be introduced as dopant? Please be specific.

8) Is there any information in literature on the transport properties of CoSb2 (the secondary phase)? This is relevant for the interpretation. Please carefully explore the literature and discuss it in the manuscript.

9) Table 1 is missing in the manuscript. Only the captions are provided.

10) The authors argue on page 4 that CoSb2 hinders the growth of Co4Sb12 grains, but this is not supported by literature or any additional data. Please elaborate.

11) Roughness can affect EDS mapping provided in Fig. 4. The observed elemental segregations can be an artifact. Please discuss this issue.

12) Please provide a reference for the Pisarenko relation.

English is satisfactory.

Round 2

Reviewer 1 Report

Accepted

Accepted

Author Response

Thank you for your acceptance.

Reviewer 3 Report

The authors have responded to all of the concerns.

Author Response

Thank you for your acceptance.

Reviewer 4 Report

Binh and coworkers have improved the manuscript to some extent, but there are still several unresolved issues, as listed below. The manuscript is worth publishing, but the authors should still improve it.

1) The transport properties in Table 1 are listed with too many digits. No theoretical or experimental studies are this precise. Please reduce the digits and show only the relevant ones.

2) The motivation is still not clear (introduction). Why studying Ce in this particular SKD? What are open questions to be answered in this work? The authors should supply all information in the manuscript and not only in the rebuttal.

3) The authors should quantify impurities in the samples? How much oxygen is incorporated? Supplying the raw EDS data doesn’t really help a reader. Please also number the figures in the supplementary information and provide captions as well as short descriptions.

4) There are typos in the manuscript (e.g., it should be “Lorenz” and not “Lorent” in the supplementary information – the y axis of the last figure).

5) Formatting of the references is very strange. All journal names are abbreviated with single letters (e.g., “Applied Thermal Engineering” (Ref 1) has been shorted into “A.t.e.” and even merged with the initials of the last author yielding “X.J.A.t.e.”). How is this compatible with the MDPI style or any other standard abbreviations? Please follow the correct style for all references.

6) Comparison with literature is still missing. The authors have provided some numerical comparisons and there are no attempts at all to explain the differences. Why is ZT lower or higher? Similar goes for other discussions.

7) Experimental details are still missing. What is the heat capacity of the compounds synthesized in this study? Please provide the data required to reproduce the results obtained herein.

8) The authors are inconsistent with the use of symbols (“k” vs. “K” for the thermal conductivity, see page 2).

English is satisfactory.
